# Associations of Occupational Heat Stress and Noise Exposure with Carotid Atherosclerosis among Chinese Steelworkers: A Cross-Sectional Survey

**DOI:** 10.3390/ijerph19010024

**Published:** 2021-12-21

**Authors:** Lihua Wang, Miao Yu, Shengkui Zhang, Xiaoming Li, Juxiang Yuan

**Affiliations:** Department of Epidemiology and Health Statistics, School of Public Health, North China University of Science and Technology, Tangshan 063210, China; wanglihua@stu.ncst.edu.cn (L.W.); yumiao@stu.ncst.edu.cn (M.Y.); zhangsk@stu.ncst.edu.cn (S.Z.); lixiaoming@ncst.edu.cn (X.L.)

**Keywords:** occupational heat stress, occupational noise exposure, carotid plaque, atherosclerosis, steelworkers

## Abstract

Occupational exposure to heat stress and noise at the workplace are widespread physical hazards and have been associated with an increase in both morbidity and mortality. This study aims to examine the association between occupational heat stress and noise exposure and carotid atherosclerosis in Chinese steelworkers. A total of 3471 subjects were included in this study. Carotid plaque was measured using ultrasonography. The occupational information was collected by face-to-face personal interviews and all of the reported information was verified with the company’s records. Workers were divided into non-exposure and exposure groups according to the company’s records regarding previous and/or current heat stress and noise exposure status in the workplace. The prevalence of carotid plaque was 30.1% in the study population and workers exposed to both occupational heat stress and noise had the highest prevalence of carotid plaque at 37.2%. The odds of carotid plaque in individuals of different exposure status were significantly elevated after adjustment for potential confounders, especially in the heat stress and noise exposure combination group: OR = 1.32, 95% CI: 1.06 to 1.65, in individuals who had experienced heat stress exposure; OR = 1.49, 95% CI: 1.18 to 1.88, in individuals who had experienced noise exposure; OR = 2.02, 95% CI: 1.60 to 2.56, in the combination group. No significant association in female workers and no significant multiplicative or additive interactions were found between occupational heat stress and noise exposure and carotid plaque. Exposure to occupational heat stress and noise are statistically associated with carotid atherosclerosis among male steelworkers.

## 1. Introduction

Occupational exposure to excessive heat in the workplace is a well-known hazard and has been associated with an increase in both morbidity and mortality [1]. Many occupations, disease states, as well as stages of life are especially vulnerable to the stress imposed on the cardiovascular system during exposure to hot ambient conditions [2]. Noise has been identified as one of the most widespread physical hazards in occupational health and safety, with 600 million persons estimated to be working in environments with hazardous levels of noise [3,4]. Heat stress and noise are considered the prime factors affecting productivity/performance of workers among occupational hazards [5].

Persistent exposure to occupational or environmental noise and heat may induce a variety of adverse health effects, such as cardiovascular autonomic function, hypertension and immunological parameters [3,6]. It has been shown that chronic exposure to noise in the workplace is associated with an approximately three fold rise in the prevalence of angina pectoris and a two fold increase in the prevalence of coronary heart disease (CHD) and isolated diastolic hypertension, especially in young workers [7], as well as sleeping disorders and psychological stress [8]. Workers of steel plants who are exposed to high temperatures showed a prevalence of metabolic syndrome and reported a significantly higher abnormal rate of electrocardiogram (ECG) [9]. Altered autonomic activity and exposure to high temperatures for a varied duration of exposure will most likely multiply the cardiovascular risk. However, most studies of the cardiovascular effects of these exposures in humans have focused on their association with blood pressure. Zhou et al. observed that noise-induced hearing loss and cumulative noise exposure time were positively correlated with blood pressure [3]. However, Tessier-Sherman et al. found that the adjusted hazard ratio of incident hypertension did not significantly differ between groups by a cumulative continuous or categorized noise exposure metric [10]. Inconsistencies between studies may be due to different study designs and important covariate adjustments, such as hearing protection devices. Other cardiovascular end points remain largely unexplored.

In addition, workers often operate in environments where multiple occupational exposures exist simultaneously, although few studies have considered them simultaneously. The effect of independent and combined physical hazardous agents on health is varied. Combined exposure to noise and heat has more of a detrimental effect on salivary cortisol and blood pressure as compared to independent exposure [11], while another study showed that subtle changes of blood pressure were traced when exposed to a combination of heat and noise [12]. Several underlying mechanisms have been shown to be likely involved in the development of these exposures to atherosclerosis: heat stress can increase the production of reactive oxygen species (ROS), induce apoptosis, and increase peripheral circulation and sweating, which could increase the plasma viscosity and serum cholesterol level, and promote the genesis and development of atherosclerosis [13,14,15,16]. Noise causes oxidative stress, inflammatory processes and a pro-thrombotic phenotype, which in turn impacts negatively on the vascular system, resulting in endothelial dysfunction and an increased risk of atherosclerosis [17,18].

Cardiovascular diseases (CVD) continues to be the leading cause of death and the largest contributor to premature mortality worldwide [19], being the cause of 40% of deaths in the Chinese population [20]. Studies in China’s workplaces show that malignant neoplasm, cardiovascular diseases, respiratory diseases, infectious diseases were the first four illnesses that threaten workers’ lives, and they accounted for 73.2% of all deaths [21]; overweight or obesity, hypertension, coronary heart disease and stroke are the main causes of workers’ cardiovascular deaths [22,23]. In fact, CVD develops over a long period of time with physical changes beginning decades before the disease manifests itself. However, some of the health hazards of occupational exposures may remain undetected when a disease condition is used as an endpoint, as there is a subsequent difficulty concerning the healthy worker effect as workers who cannot tolerate the working environment may change to other work when the first symptoms of CVD occur. Therefore, it is important to include surrogate parameters that describe early subclinical changes [24]. Ultrasound examination of the carotid arteries is an easily accessible tool by which one can identify early subclinical atherosclerosis by the detection of plaques. The presence of carotid plaques has been shown to improve significantly the better [25] risk prediction of major cardiovascular events [26], and the accuracy for the diagnosis of coronary artery disease [27].

To our knowledge, no previous study has examined the association between occupational heat stress and noise exposure with carotid plaque examined by imaging techniques. Based on the foregoing, in this cross-sectional study, we examined whether long-term exposure to occupational heat stress and noise were associated with carotid atherosclerosis among steelworkers.

## 2. Materials and Methods

### 2.1. Study Design and Population

This study was based on cross-sectional data from the occupational population, which was conducted among steelworkers at 11 steel production departments owned by the HBIS Group’s Tangsteel Company in Tangshan City, Hebei Province in north China. All workers at this company underwent a legally required health examination each year. A total of 7661 participants who underwent the annual legally required occupational health examinations were recruited from February to June 2017. There were 4084 workers who volunteered and completed carotid ultrasound examinations. After excluding 205 workers without sufficient shift work data, 297 workers without complete information on main covariates on the questionnaire, 111 workers without occupational heat stress and noise data, a total of 3471 participants were included for the final analysis. All participants gave informed consent before taking part in this study. The research was approved by the Ethics Committee of North China University of Science and Technology (No. 16040).

### 2.2. Measurement of Plaque in the Carotid Artery

Assessment of plaque from both the left and right carotid artery systems was performed using a high-resolution B-mode topographic ultrasound system (PHILIPS, HD7, China) by two trained sonographers who were blinded to the research purpose and the study design. Participants were examined in the supine position with their head rotated in the opposite direction to the probe and with a lateral probe orientation.

For this study, atherosclerotic plaques were defined as focal structures encroaching into the arterial lumen of at least 0.5 mm or 50% of the surrounding intima-media thickness (IMT) value, or demonstrates a thickness >1.5 mm as measured from the intima-lumen interface to the media-adventitia interface [28]. When a local protrusion was defined as a plaque, its maximum thickness (mm) was measured using ultrasound calipers. The carotid plaque score indicates the severity of atherosclerosis, which is the sum of the cumulative maximum thickness of plaques obtained in the longitudinal sections of the common carotid artery, bifurcation, and internal carotid artery of the left and right carotid systems [29].

### 2.3. Assessment of Occupational Exposure

All of the participants were interviewed face-to-face by trained investigators of the research team following a structured questionnaire. After checking the completeness and correctness of each questionnaire, we used a customized program to scan and transform the handwritten data into an electronic data set immediately. The transformed version of each questionnaire was checked by two skilled investigators separately, and the checked results were then compared and reviewed before the final submission. The questionnaire mainly includes the following parts: socio-demographic characteristics, job characteristics, personal characteristics, living and behavioral habits. For job characteristics, the participants recruited were asked to report whether they were exposed to occupational heat stress and/or noise in their work environment, and if yes, they were further asked if they had cooling protections and/or wore earplugs. All of this reported information was verified with the company’s records (basic job information including workshop, type of work and type of occupational hazard exposure). Anthropometric and blood biochemical indicators of each worker were collected on the same day.

The occupational hazard factors were measured in June 2017 by a qualified third-party company in accordance with the National Occupational Health Standards of the People’s Republic of China. Exposure to heat stress work (high temperature) was defined as the average wet-bulb globe temperature (WBGT) index of the workplace is equal or greater than 25 °C in the process of production (GBZ 2.2–2007) [30]. The WBGT index was measured by the WBGT-2006 high temperature index instrument (SW23-02). If there was no productive heat source in the workplace, three measuring points were selected to take the average value of WBGT index. Where there was a productive heat source, three to five measuring points were selected to take the average value of the WBGT index. If the workplace was isolated into different thermal or ventilated environment, two measuring points were selected to take the average value of WBGT index (GBZ/T 189.7–2007) [31]. Exposure to noise was defined as workers who were exposed to a noisy environment where the 8 h/d or 40 h/week equivalent A-weighted sound pressure level is ≥80 dB, which may be harmful to health and hearing (GBZ/T 229.4–2012) [32]. The workplace production noise was measured by the HS6288B Noise Spectrum Analyzer (SW19-01). If the distribution of the sound field in the workplace was uniform (between-field difference of A-sound levels were less than 3 dB(A)), three measuring points were selected to obtain the average value, otherwise the sound field should be divided into several sound level areas. In each sound field, two measuring points were selected to take the average value (GBZ/T 189.8–2007) [33]. Workers were divided into non-exposure and exposure groups according to the company’s records concerning previous and/or current heat stress and noise exposure status in the workplace.

The main work schedule of the present study population has been introduced in detail in our previous research [34]. In brief, shift work in this study refers to rotating night shifts (the mainly four-crew-three-shift system now and historical three-crew-two-shift system) and was divided into never, ever and current, according to the current shift status.

### 2.4. Assessment of Covariates

Covariates mainly included age, sex, marital status, educational level, shift work, cooling protection, wear earplugs and established risk factors for CVD [35]: body mass index (BMI), smoking status, drinking status, diet (dietary approaches to stop hypertension, DASH), physical activity, sleep duration, insomnia (AIS score), hypertension, diabetes, dyslipidemia, the use of antihypertensive, antidiabetic and lipid-lowering drugs. Occupational covariates include shift work, cooling protection and wear earplugs (see Appendix A).

### 2.5. Statistical Analysis

Continuous variables are presented as means and standard deviations, and between-group comparisons were performed using Student’s *t*-test or analysis of variance (ANOVA) of normally distributed data. Categorical variables are presented as numbers and percentages, and the chi-square test was used to compare differences among groups. Associations between occupational heat stress and noise exposure and carotid plaque were reported as odds ratios (OR) and corresponding 95% confidence intervals from multivariable adjusted logistic regression models. The risk factors and potential confounders were included in the analysis. We fit a multivariable model (model 1) including age and sex, and a model (model 2) additionally adjusted for other confounders and known risk factors (marital status, educational level, BMI, smoking status, drinking status, DASH score, physical activity, sleep duration, insomnia, hypertension, diabetes, dyslipidemia, medication use), and a final model (model 3) additionally adjusted for occupational factors (shift work, cooling protection and wear earplugs). In subgroup analysis, we introduced the stratifying factors, including sex, BMI (<25 kg/m^2^ or ≥25 kg/m^2^) and shift work (never, ever, current) to assess potential effects modification. The comprehensive test, goodness of fit test and multicollinearity test of all of the models were evaluated. The Box-Tidwell method was used to test whether there was a linear relationship between the continuous independent variables and the logit transformation value of the dependent variables. A two-tailed *p* < 0.05 was considered statistically significant. All statistical analyses were performed using SAS V.9.4 (SAS Institute, Cary, NC, USA).

## 3. Results

### 3.1. General Characteristics of the Participants

The present study of 3471 included participants consisted of 90.4% males, with a mean age of 46.1 years, and a mean BMI of 25.2 kg/m^2^. The prevalence of hypertension, diabetes and dyslipidemia in the study participants were 32.5%, 13.5%, and 40.3%, respectively. Table 1 shows the basic characteristics of the participants according to the carotid plaque status. Current smoking, current drinking and insomnia were more likely to be reported among workers with plaque. Workers with plaque had higher rates of shift work. As shown in Appendix A, the prevalence of carotid plaque in males and females were 32.0% and 12.4%, respectively. Compared with female workers, male workers had higher proportions of smoking, drinking, overweight, obesity, hypertension, diabetes, and dyslipidemia. Appendix A shows the general characteristics of the study participants according to the occupational exposure status. Workers exposed to heat stress and noise simultaneously had shorter sleep duration and higher rates of shift work, overweight, obesity, smoking, drinking and dyslipidemia. Workers exposed to both heat stress and noise had a higher prevalence of carotid plaque than those exposed to only one condition.

### 3.2. Association between Occupational Exposure and Carotid Plaque

The prevalence of carotid plaque was 30.1% overall in this study population, and workers exposed to both heat stress and noise had the highest prevalence of carotid plaque at 37.2% (Appendix A). Table 2 shows results from the logistic regression model. Compared with non-exposure workers, significantly increased odds ratios (OR) of carotid plaque were observed in each exposure status, either alone or in combination with heat stress and noise, after adjusting for age and sex (Model 1). After additionally adjusting for marital status, educational level, BMI, smoking status, drinking status, DASH score, physical activity, sleep duration, insomnia, hypertension, diabetes, dyslipidemia and medication use, this association remained robust (Model 2). After additionally adjusting for shift work, cooling protection and whether or not earplugs were used, the odds of carotid plaque in each exposure status were slightly attenuated but remained significantly elevated, especially in the combination group (OR = 1.32, 95% CI: 1.06 to 1.65, in heat stress exposure; OR = 1.49, 95% CI: 1.18 to 1.88, in noise exposure; OR = 2.02, 95% CI: 1.60 to 2.56, in both heat stress and noise exposure; Model 3). However, no significant multiplicative or additive interactions between the heat stress and noise on the odds of carotid plaque were observed (Appendix A).

We analyzed the relationship between occupational exposures and carotid plaque through stratification analysis based on potential effect modifiers (Table 3). Compared with non-exposure workers, elevated odds of carotid plaque were observed in each exposure status in most subgroup analyses. Stratifying by sex showed significant estimates for male but not for female smokers: OR = 1.33, 95% CI: 1.06 to 1.68, in heat exposure; OR = 1.55, 95% CI: 1.21 to 1.97, in noise exposure; OR = 2.15, 95% CI: 1.69 to 2.74, in both heat and noise exposure; in males. When stratifying by BMI, ORs were all significant for workers in two subgroups. Stratifying by shift work showed significant estimates for ever/current but not for never shift workers: OR = 1.31, 95% CI: 1.02 to 1.67, in heat exposure; OR = 1.47, 95% CI: 1.13 to 1.91, in noise exposure; OR = 2.04, 95% CI: 1.60 to 2.61, in both heat and noise exposure; in ever/current shift work. There was no significant effect modification of the association between occupational exposures and carotid plaque by BMI and shift work (all *p* for interaction term of exposure vs. non-exposure > 0.05).

In order to evaluate whether the association between occupational exposures and carotid plaque was generally impacted by decades of aging, by the degree of smoking and drinking, we adjusted the age group and detailed classification of smoking and drinking in logistic regression models (other variables are the same as Model 3 in Table 3), and the results were diluted but remained significant (Appendix A).

## 4. Discussion

In this cross-sectional study of occupational populations, we examined the association between the occupational exposure of heat stress and noise and carotid plaque. Positive associations were observed between different exposure statuses and the odds of carotid plaque, adding evidence to an underlying pro-atherogenic role of occupational heat stress and noise exposure in cardiovascular disease. The present study also provided additional evidence concerning relationships between exposure heat stress and noise simultaneously and carotid plaque among steelworkers, which have never been reported in previous studies.

Our findings are largely consistent with previous studies showing a positive association of heat stress and noise with adverse health effects. A systematic review and meta-analysis from 111 studies done in 30 countries, including 447 million workers from more than 40 different occupations, revealed that 15% studies of individuals who typically or frequently worked under heat stress (minimum of 6 h per day, five days per week, for two months of the year) experienced kidney disease or acute kidney injury [36]. Nonlinear dose-response relationship between cumulative high temperature exposure [37] and cumulative noise exposure [38] and hypertension were found after adjusting for confounding factors using a restricted cubic spline model. Analyses on the baseline data of the Dongfeng-Tongji Cohort Study identified occupational noise exposure as a potential risk factor for increased hypertension risk [39]. A cross-sectional survey revealed self-reported noise exposure and audiometrically measured hearing loss were strongly associated with the prevalence of hypertension in steelworkers [18]. The Whitehall II cohort study of British civil servants revealed that long-term exposure to night-time road traffic noise was associated with higher carotid IMT [40]. However, the extent and consequences of heat stress exposure in different occupational settings, countries, and cultural contexts are not well studied [41], and there is limited research on the effects of chronic occupational heat stress and noise exposure on carotid plaque.

It is noteworthy that these studies above only consider single exposures, but actual production environments often have multiple exposures at the same time. An experimental study revealed that the average saliva cortisol and blood pressure in male and female subjects increased significantly after independent exposure to noise at 95 dB(A) (other levels of 45, 75, 85 dB(A)) and a wet bulb globe temperatures (WBGT) index of 34 °C (other levels of 22 and 29 °C). The combined exposure to noise and heat increased saliva cortisol and blood pressure, which was statistically significant for three combinations of 95 dB(A) at 34 °C, 95 dB(A) at 29 °C, and 85 dB(A) at 34 °C [11]. Another experimental study performed in a climatic chamber observed that the mean change of systolic blood pressure was different significantly before and after exposure to heat and noise levels including 75, 85, and 95 dB (*p* = 0.015, *p* = 0.001, *p* > 0.001, *p* = 0.027, respectively). Although systolic and diastolic blood pressures changed drastically, it was not significantly different in simultaneous exposure to heat and noise [12]. Despite the odds of carotid plaque being higher in the combination group than either single group in our study, no significant multiplicative or additive interactions between the heat stress and noise were observed. However, the effects of combined exposure to physical hazardous agents are not always greater than the influence of single exposure; rather it depends on the type of exposure to these physical agents. When various environmental hazardous agents are involved together, if the effects of combined exposure are more pronounced than the effects of independent exposure, they reinforce each other’s effects and have a similar influential mechanism; however, if the effects of combined exposure are equal to the effects of an independent exposure, they are likely to have different influential mechanisms [11]. Considering the fact that exposure to these environmental factors are combined in most working environments [42], more attention should be paid to the effects of combined exposure on these physical hazardous agents.

It should be noted that cooling protection and wearing earplugs had little effect on the association between occupational heat stress and noise. Our study used a dichotomous yes/no response for the cooling protection and wearing earplugs status. When the categories of responses were presented in terms of frequency (e.g., always, often, sometimes, rarely, never), the dichotomy collapsed the latter three categories to indicate routine nonuse of personal protective equipment (PPE). Hessel’s study suggested that 24% to 27% Albertans’ electricians, plumbers and pipefitters reported “never, seldom or sometimes” wearing hearing protection (HP) [43]. In a Canadian study, 20% workers in the workplace indicated that they “sometimes”, “rarely”, or “never” wore HP, and HP was often not worn consistently even when occupationally required [44]. The causes of irregular PPE usage include hindrance to communication, discomfort, perceived self-efficacy, and perceived benefits or barriers. The frequency and duration of PPE use on workers’ potential health and safety may be worth exploring in future studies.

The present study showed an association between both occupational heat stress and noise exposure and carotid plaque among male workers, but not among female workers. As females usually develop arterial diseases approximately one decade later than males—which is related to a female’s estrogen production—it is possible that the effects of occupational exposure on atherosclerosis appear later in females than in males [45]. Moreover, compared with male workers, a higher proportion of female workers were not exposed to occupational heat stress and noise and did not work shifts, and had a healthier lifestyle. In the same exposure environment, the exposure duration and level of female workers were relatively low due to different job categories. Since female workers over the age of 55 enter retirement; the female workers were younger and the number was relatively small in our study. Hence, the association between occupational exposures and atherosclerotic process in females should be further explored in large-scale prospective studies. Our study showed an association between both occupational heat stress and noise exposure and carotid plaque only among ever/current workers. Due to the widespread shift system in steel plants, workers are exposed to heat stress and noise at night almost a quarter of the time, which provides a further direction for exploring the effect of cumulative heat stress and noise exposure at night on CVD. This implies that reducing worker’s shifts can reduce the risk of subclinical atherosclerosis in workers exposed to occupational heat stress and noise so as to protect workers’ health.

The major strengths of our study involved the large sample size, accurate measurement of carotid plaque by ultrasonography, and a comprehensive range of potential confounders to more precisely study the effect of occupational exposures on carotid plaque. However, our research also has certain limitations. First, the cross-sectional design restricts the evidence for causal inferences between occupational exposures and carotid plaque. Second, occupational heat stress and noise exposures were not measured at individual levels and there was a lack of quantitative measurements (e.g., intensity, stimulus duration, frequency content) which accurately reflect an individual’s actual exposure. In addition, our questionnaire lacked information on how often and how long workers use cooling protection and wear earplugs at work, which could affect the extent of association between exposure and outcome. Third, we had no other noise source included with a similar mean sound pressure level (e.g., leisure time noise exposure, environmental noise and road traffic noise) to elucidate whether occupational noise has a specific and unique impact on the carotid atherosclerosis. Fourth, our study participants are all front-line workers from the production sector, so it was not possible to take into account the occupational category (office or physical workers) that could be confounder of atherosclerosis presence. Fifth, those who are competent for long-duration occupational exposures are more likely to have better physical fitness (healthy worker effect) or have acclimated to them, which may result in an underestimation of the association between the exposure and outcome. Finally, as our study was conducted in the steel production occupational setting where the vast majority of steelworkers are male workers, generalization to other populations remains limited. Considering the above limitations, large-scale prospective studies are needed in the future. It is necessary to increase the data collection of exposure sources other than in the occupational environment (such as road traffic noise and meteorological high temperature), adopt individual sampling equipment to accurately collect workers’ exposure intensity and the amount of occupational harmful factors, and ask workers about the types of PPE used and the extent of their donning.

## 5. Conclusions

Exposure to occupational heat stress and noise were statistically associated with carotid atherosclerosis among male steelworkers. Further large-scale prospective longitudinal studies are warranted to confirm our findings.

## Figures and Tables

**Table 1 ijerph-19-00024-t001:** Basic characteristics of participants according to carotid plaque.

Variables	Total	Without Plaque	With Plaque	*p*-Value
	*n* = 3471	*n* = 2426	*n* = 1045	
Sex (male), *n* (%)	3139 (90.4)	2135 (88.0)	1004 (96.1)	<0.001
Age (years), mean (SD)	46.1 (7.8)	44.4 (8.0)	49.9 (5.9)	<0.001
Sleep duration (h), mean (SD)	6.75 (1.21)	6.79 (1.20)	6.66 (1.22)	0.006
DASH score, mean (SD)	21.6 (2.4)	21.6 (2.3)	21.6 (2.5)	0.796
BMI (kg/m^2^), mean (SD)	25.2 (3.3)	25.1 (3.3)	25.5 (3.2)	0.003
Systolic blood pressure (mmHg), mean (SD)	129.6 (16.6)	127.2 (15.7)	135.4 (17.1)	<0.001
Diastolic blood pressure (mmHg), mean (SD)	82.9 (10.6)	81.6 (10.3)	85.9 (10.8)	<0.001
Fasting blood glucose (mmol/L), mean (SD)	6.13 (1.37)	5.99 (1.17)	6.45 (1.72)	<0.001
Total cholesterol (mmol/L), mean (SD)	5.15 (0.99)	5.02 (0.94)	5.43 (1.03)	<0.001
Triglycerides (mmol/L), mean (SD)	1.69 (1.56)	1.68 (1.61)	1.73 (1.44)	0.414
HDL-C (mmol/L), mean (SD)	1.31 (0.33)	1.30 (0.32)	1.33 (0.34)	0.027
LDL-C (mmol/L), mean (SD)	3.25 (0.87)	3.15 (0.83)	3.50 (0.93)	<0.001
Age (years), *n* (%)				<0.001
23–29	144 (4.2)	142 (5.9)	2 (0.2)	
30–39	569 (16.4)	497 (20.5)	72 (6.9)	
40–49	1427 (41.1)	162 (43.8)	365 (34.9)	
50–60	1331 (38.4)	725 (29.9)	606 (58.0)	
Marital status, *n* (%)				<0.001
Unmarried	89 (2.6)	83 (3.4)	6 (0.6)	
Married	3288 (94.7)	2275 (93.8)	1013 (96.9)	
Other	94 (2.7)	68 (2.8)	26 (2.5)	
Education level, *n* (%)				<0.001
Primary or Middle	1027 (29.6)	599 (24.7)	428 (41.0)	
High school or college	1846 (53.2)	1327 (54.7)	519 (49.7)	
University and above	598 (17.2)	500 (20.6)	98 (9.4)	
Smoking status, *n* (%)				<0.001
Never	1435 (41.3)	1093 (45.1)	342 (32.7)	
Ever	233 (6.7)	156 (6.4)	77 (7.4)	
Current (<11 pack–years)	657 (18.9)	471 (19.4)	186 (17.8)	
Current (11–20 pack–years)	352 (10.1)	248 (10.2)	104 (10.0)	
Current (>20 pack–years)	794 (22.9)	458 (18.9)	336 (32.2)	
Drinking status, *n* (%)				<0.001
Never	2014 (58.0)	1525 (62.9)	489 (46.8)	
Ever	118 (3.4)	69 (2.8)	49 (4.7)	
Current (<140 g/week)	986 (28.4)	641 (26.4)	345 (33.0)	
Current (≥140 g/week)	353 (10.2)	191 (7.9)	162 (15.5)	
Physical activity, *n* (%)				0.743
Low	37 (1.0)	28 (1.2)	9 (0.9)	
Moderate	246 (7.1)	172 (7.1)	74 (7.1)	
High	3188 (91.9)	2226 (91.8)	962 (92.1)	
Sleep duration (h), *n* (%)				0.127
<6	433 (12.5)	289 (11.9)	144 (13.8)	
≥6	3038 (87.5)	2137 (88.1)	901 (86.2)	
Insomnia, *n* (%)				<0.001
No	2264 (65.2)	1635 (67.4)	629 (60.2)	
Yes	1207 (34.8)	791 (32.6)	416 (39.8)	
BMI (kg/m^2^), *n* (%)				0.174
<25	1732 (49.9)	1235 (50.9)	497 (47.6)	
25–30	1467 (42.3)	1008 (41.6)	459 (43.9)	
≥30	272 (7.8)	183 (7.5)	89 (8.5)	
Hypertension, *n* (%)				<0.001
No	2344 (67.5)	1758 (72.5)	586 (56.1)	
Yes	1127 (32.5)	668 (27.5)	459 (43.9)	
Diabetes, *n* (%)				<0.001
No	3002 (86.5)	2173 (89.6)	829 (79.3)	
Yes	469 (13.5)	253 (10.4)	216 (20.7)	
Dyslipidemia, *n* (%)				<0.001
No	2072 (59.7)	1525 (62.9)	547 (52.3)	
Yes	1399 (40.3)	901 (37.1)	498 (47.7)	
Drug use (yes), *n* (%)	314 (9.1)	153 (6.3)	161 (15.4)	<0.001
Shift work, *n*(%)				<0.001
Never	679 (19.6)	534 (22.0)	145 (13.9)	
Ever	730 (21.0)	514 (21.2)	216 (20.7)	
Current	2062 (59.4)	1378 (56.8)	684 (65.5)	
Cooling protection, *n* (%)	2600 (74.9)	1818 (74.9)	782 (74.8)	0.948
Wear earplugs, *n* (%)	1653 (47.6)	1140 (47.0)	513 (49.1)	0.256
Exposure status, *n* (%)				<0.001
Non-exposure	1036 (29.9)	768 (31.7)	268 (25.7)	
Heat stress only	885 (25.5)	622 (25.6)	263 (25.2)	
Noise only	797 (23.0)	564 (23.3)	233 (22.3)	
Heat stress + Noise	753 (21.7)	472 (19.5)	281 (26.9)	

Values are expressed as the mean (SD) or number (%); *p* values were from Pearson’s χ^2^ test for categorical variables and *t*-test for continuous variables. DASH, dietary approaches to stop hypertension; BMI, body mass index; HDL-C, high density lipoprotein cholesterol; LDL-C, low density lipoprotein cholesterol.

**Table 2 ijerph-19-00024-t002:** Multivariate adjusted ORs between carotid plaque and different occupational exposure status.

Exposure Status	Model 1	Model 2	Model 3
Non-exposure	1.00	1.00	1.00
Heat stress only	1.24 (1.00 to 1.53)	1.33 (1.07 to 1.66)	1.32 (1.06 to 1.65)
Noise only	1.46 (1.17 to 1.83)	1.53 (1.22 to 1.93)	1.49 (1.18 to 1.88)
Heat stress + Noise	2.10 (1.69 to 2.62)	2.27 (1.81 to 2.85)	2.02 (1.60 to 2.56)

Model 1: Adjusted for age (continuous variable), sex (male, female). Model 2: Additionally adjusted for marital status (unmarried, married, other), educational level (primary or middle, high school or college, university and above), BMI (<25 kg/m^2^, 25–30 kg/m^2^, ≥30 kg/m^2^), smoking status (no/yes), drinking status (no/yes), DASH score (continuous variable), physical activity (low, moderate, high), sleep duration (<6 h, ≥6 h), insomnia (no/yes), hypertension (no/yes), diabetes (no/yes), dyslipidemia (no/yes) and medication use (no/yes) on Model 1. Model 3: Additionally adjusted for shift work (never, ever, current), cooling protection (no/yes) and wear earplugs (no/yes) on Model 2. The Omnibus Tests of Model Coefficients of the three models were generally meaningful with all *p* < 0.01, the *p* values of Hosmer and Lemeshow tests for the three models were 0.641, 0.193 and 0.863, respectively, indicating a high goodness of fit. The variance inflation factor (VIF) of the covariates in the three models ranged from 1.004 to 1.489, without multicollinearity.

**Table 3 ijerph-19-00024-t003:** Multivariate adjusted ORs between carotid plaque and different exposure status by sex, BMI and shift work.

Groups	Non-Exposure	Heat Stress Only	Noise Only	Heat Stress + Noise
Sex				
Male	1.00	1.33 (1.06 to 1.68)	1.55 (1.21 to 1.97)	2.15 (1.69 to 2.74)
Female	1.00	1.34 (0.48 to 3.74)	0.85 (0.34 to 2.14)	0.19 (0.03 to 1.24)
BMI				
<25 kg/m^2^	1.00	1.40 (1.01 to 1.94)	1.57 (1.12 to 2.20)	2.17 (1.54 to 3.07)
≥25 kg/m^2^	1.00	1.21 (0.89 to 1.64)	1.44 (1.03 to 2.00)	1.98 (1.43 to 2.74)
Shift work				
Never	1.00	1.34 (0.79 to 2.27)	1.56 (0.92 to 2.65)	1.78 (0.78 to 4.05)
Ever/current	1.00	1.31 (1.02 to 1.67)	1.47 (1.13 to 1.91)	2.04 (1.60 to 2.61)

Adjusted for age (continuous variable), sex (male, female), marital status (unmarried, married, other), educational level (primary or middle, high school or college, university and above), BMI (<25 kg/m^2^, 25–30 kg/m^2^, ≥30 kg/m^2^), smoking status (no/yes), drinking status (no/yes), DASH score (continuous variable), physical activity (low, moderate, high), sleep duration (<6 h, ≥6 h), insomnia (no/yes), hypertension (no/yes), diabetes (no/yes), dyslipidemia (no/yes), medication use (no/yes), shift work (never, ever, current), cooling protection (no/yes), wear earplugs (no/yes). The Omnibus Tests of Model Coefficients of the six models were generally meaningful with all *p* < 0.01, the *p* values of Hosmer and Lemeshow tests for the six models were ranged from 0.176 to 0.794, indicating a high goodness of fit. The variance inflation factor (VIF) of the covariates in the six models ranged from 1.006 to 1.483, without multicollinearity.

## Data Availability

The data presented in this study are available on request from the corresponding author.

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
