# Peer review of "Associations of Occupational Heat Stress and Noise Exposure with Carotid Atherosclerosis among Chinese Steelworkers: A Cross-Sectional Survey"

_ijerph, 2021, doi:10.3390/ijerph19010024_

Round 1

Reviewer 1 Report

To editor,

I have revised the article “Associations of occupational heat stress and noise exposure with carotid atherosclerosis among Chinese steelworkers a cross-sectional survey” follow my observations.

This is an interesting topic and the authors managed to mount a quite large sample of subjects, which is already impressive. However, there is some methodological issues that should be addressed. First, the heat and sound stress are categorical. But, physiologically those two effects are highly non linear. This raises the question of the usefulness of having multiples classes.

There is also issue with the number presented in the analyse at the line 86 there is 3582 participants, but 3471 at the line 162.  Also, many p-values of the table 1 do not make sense and are inconsistent with the supplemental material. This raises serious suspicion about the data analysis. What is the signification of the p-value in the table S2

Also, I am very surprised that cooling protection and wear earplugs have such a little effect. This certainly need more investigation and discussion. As for model presentation, significance of all model parameters should be given in the supplementary material and most significant variable should be discussed in the text.

Overall, I suggest this text should have major revision before publication.

In addition, a noted a typo at the line 156 : 25 kg/m2.

Reviewer 2 Report

While mentioned in passing, the difference between association (correlation) and causation should be more clearly and directly stated. Yes, there is an association between heat stress and noise with atherosclerosis...but there was not adequate evidence presented to show causation. This is an important factor that should be emphasized.

Reviewer 3 Report

Thanks for the opportunity to read your interesting study. Please consider my comments below to improve the quality of your paper.

Introduction
-General comment: The structure of this section reads a bit confusing. Instead of going forth and back to different concepts, I suggest you follow a more consistent discussion of the literature. I recommend mentioning first the prevalence and effects of noise and heat in the work environment and then moving to the CVD area.
-Lines 33-35. Only one study per argument is not adequate to claim the associations. Would you please add 2-3 more sources per argument?
-Line 37: Citation No 3 refers only to heat, not noise.
-When discussing noise and heat in this section, you must also present other studies revealing their synergetic effects.
-Lines 41-43: The literature sources regard the whole population. Can you add and discuss a few sources focusing on workers?
-Lines 54-55: Mention indicative effects.
-Lines 61, 97; Expand the acronyms
-Lines 63-64; Explain with a few words the inconsistency
-Lines 68-69; Give more details/some examples of those underlying mechanisms.

Section 2
-Lines 105-106
--Be more specific upfront about the information you collected. What did you ask and why?
--Do you mean you conducted 3582 interviews? Was the questionnaire mentioned in section 2.4 filled during those sessions? Was the health examination also performed concurrently? Please state your methods in one place and then explain each of them, if applicable.
--How did you verify interview data against company records? This remains unclear as you have not explained what information you collected during the interviews.
-Lines 112-127: Did you take measurements during the study? Where and how many each time? Or are you referring to measurements the company made? It is unclear.
-Section 4.2 and supplementary file:
--Smoking status reads vague and not the same clear as drinking status.
--What does 'ever' mean for smoking and drinking? Do you mean they smoked/drank in the past, but they have stopped?
--Why haven't you considered the degree of exposure to smoking and alcohol? It is not the same to smoke half a pack with a whole pack, for example.

Section 3.1
It would help if you redid your analysis by considering the following fundamental statistics principles collectively:
-You have inflated the error rate by performing multiple tests on the same dependent data. You must either apply a correction or randomly stratify your sample into smaller datasets with data points enough to provide you with a good power level.
-The very large sample sizes per test have led to the detection of significant differences even in cases with identical or very close mean values. You have reported the p but not the power, which must be at least 80%. For example, if you calculate Cohen's d for Sleep Duration, you get 0% power due to the identical means. If you apply the same to Total Cholesterol, you get (5.16-5.07)/0.99 (considering the SD of the control group) = 9% power. If you want to be more precise, there are formulas to calculate the power for samples with different SDs.
-After finding the correct and proper strategy for your data analysis based on the above, revise your Tables and the respective observations/arguments accordingly.
-In addition to the above, you might want to remove some variables from your study to enable you to manage the tests and findings better. It is up to you, but in any case, be very careful with your statistical tests.

Section 3.2
I suggest you include some more information about your models. For example:
-Did you assess the models (e.g., log-likelihood statistic, deviance statistic, etc.)?
-How did you build the models with more than one predictor (e.g., forced entry)?
-Have you checked for sources of bias and other parameters (e.g., linear relationship between predictor and logit; complete separation; multicollinearity)?
-You could also report the model parameters and Wald statistics (maybe as part of the supplementary file).

Section 4
Lines 269-289 belong to the introduction, where you review the literature. In this place, they do not add anything to the discussion of your results.

Round 2

Reviewer 1 Report

The new version of this paper is much better than the original on many aspects. All my previous concerns were addressed. I suggest to publish it as is.

Author Response

Thank you very much for your valuable advice and recognition!

Reviewer 3 Report

Thanks for your efforts to address my comments and improve your manuscripts. I only have the following minor remarks for your consideration:

-Your responses to my comments on Section 3.2 of the original version of the paper must be included in the results section of the article as applicable per subsection/test. The readers must be confident you covered/checked all those aspects.

-In the conclusions mention "statistically associated" instead of simply "associated". Also, regarding future research, you need to add recommendations sourced from the limitations of your study (e.g., individual measurements, types of PPE used and extent of their donning, exposure to noise and heat beyond the work environment).

-The citation numbers must follow the order of their reference in the text. You did not attend this when reorganising the introduction section. For example, you jump from citation No 4 to 12 and later you get back to other numbers.

0Please check again for grammar and spelling errors as well as consistency. For example, in the supplementary file, you mention "drinkers" in the first paragraph whereas you obviously meant to type smokers.

Author Response

Thanks for your efforts to address my comments and improve your manuscripts. I only have the following minor remarks for your consideration:

-Your responses to my comments on Section 3.2 of the original version of the paper must be included in the results section of the article as applicable per subsection/test. The readers must be confident you covered/checked all those aspects.

Thanks so much for your suggestions. We have added the results of comprehensive tests of all regression models, goodness of fit tests, and multicollinearity tests between variables to the table notes in all results sections. (Line 330-333; Table 2 and Table 3, Line 439-442 and Line 448-451) (Supplementary file, Table S4-S6: Line 132-135, 146-148,176-178)

-In the conclusions mention "statistically associated" instead of simply "associated". Also, regarding future research, you need to add recommendations sourced from the limitations of your study (e.g., individual measurements, types of PPE used and extent of their donning, exposure to noise and heat beyond the work environment).

Thanks so much for your suggestions. The exact statistical expression has been replaced (Line 711). In addition, in view of the limitations of this study, necessary suggestions have been supplemented (Line 590-596).

-The citation numbers must follow the order of their reference in the text. You did not attend this when reorganising the introduction section. For example, you jump from citation No 4 to 12 and later you get back to other numbers.

Thanks so much for suggestions. We double-checked the correspondence between the content and the citation number to make sure it was correct.

Please check again for grammar and spelling errors as well as consistency. For example, in the supplementary file, you mention "drinkers" in the first paragraph whereas you obviously meant to type smokers.

Thank you for pointing out our carelessness. We have corrected the mistakes (Supplementary file, Line 21-22) and carefully checked the whole paper again.

Thank you once again for your comments and professional suggestions to improve the quality of our manuscript.
